# Factors associated with changes in walking performance in individuals 3 months after stroke or TIA: secondary analyses from a randomised controlled trial of SMS-delivered training instructions in Sweden

Birgit Maria Vahlberg [ID],[1] Staffan Eriksson [ID],[1,2] Ulf Holmbäck [ID],[3] Erik Lundström[4]

For numbered affiliations see end of article.

**Correspondence to**
Dr Birgit Maria Vahlberg; birgit.vahlberg@pubcare.uu.se

## ABSTRACT

**Objectives** This study aimed to identify factors related to changes in walking performance in individuals 3 months after a stroke or TIA.

**Design** Cross-sectional study with post hoc analysis of a randomised controlled study.

**Setting** University Hospital, Sweden.

**Participants** 79 individuals, 64 (10) years, 37% women, who were acutely hospitalised because of stroke or TIA between November 2016 and December 2018. Inclusion criteria were patients aged 18 or above and the major eligibility criterion was the ability to perform the 6 min walking test.

**Intervention** The intervention group received standard care plus daily mobile phone text messages (short message service) with instructions to perform regular outdoor walking and functional leg exercises in combination with step counting and training diaries. The control group received standard care.

**Outcome measures** Multivariate analysis was performed and age, sex, group allocation, comorbidity, baseline 6 min walk test, body mass index (BMI), cognition and chair-stand tests were entered as possible determinants for changes in the 6 min walk test.

**Results** Multiple regression analyses showed that age (standardised beta −0.33, 95% CI −3.8 to −1.05, p<0.001), sex (−0.24, 95% CI −66.9 to −8.0, p=0.014), no comorbidity (−0.16, 95% CI −55.5 to 5.4, p=0.11), baseline BMI (−0.29, 95% CI −8.1 to −1.6, p=0.004), baseline 6 min walk test (−0.55, 95% CI −0.5 to −0.3, p<0.001) were associated with changes in 6 min walk test 3 months after the stroke event. The regression model described 36% of the variance in changes in the 6 min walk test.

**Conclusions** Post hoc regression analyses indicated that younger age, male sex, lower BMI and shorter 6 min walk test at baseline and possible no comorbidity contributed to improvement in walking performance at 3 months in patients with a recent stroke or TIA. These factors may be important when planning secondary prevention actions.

**Trial registration number** NCT02902367.

## STRENGTHS AND LIMITATIONS OF THIS STUDY

⇒ Study data were drawn from a randomised controlled trial and we used established outcome measures.

⇒ The study design calls for precaution with causal inferences.

⇒ Our findings cannot be generalised to more disabled community-living individuals after a stroke or to individuals with chronic stroke.

⇒ The study is relatively small, making the study prone to bias, and all patients are from a single centre in Sweden, reducing the generalisability of the results.

## INTRODUCTION

Approximately three out of four stroke incidents can be attributed to behavioural risk factors—for example, unhealthy diets and low levels of physical activity.[1,2] Both stroke and TIA indicate ongoing arteriosclerotic changes in the vessels that can lead to further cardiovascular events.[1,2] Physical activity is known to decrease the risk of stroke, TIA and myocardial infarction.[3–5]

After a stroke, individuals are often predisposed to functional limitations, contributing to a further risk of recurrent stroke or cardiovascular complications.[1,6] Individuals who have suffered from a stroke take an average of 4000 steps a day in the chronic phase after stroke, which is far below the recommended 10 000 steps to meet the guidelines for physical activity.[6,7] Also, individuals with high function after a stroke limit their activity. Still, we know that even low-intensity physical activity, such as walking, is important for minimising disability and promoting long-term metabolic health.[8,9] Outdoor walking is generally easy to

perform. It is a cost-effective measure to address sedentary lifestyles and increase physical activity after stroke.

Research demonstrates that the ability to walk longer distances and good balance correlates with higher physical activity levels following a stroke.[9 10] Thus, interventions for enhancing walking performance are of crucial importance for secondary prevention.[11] Metrics such as walking distance can be used as an indicator of the level of physical activity after a stroke or TIA. In addition, understanding factors associated with improved walking is also essential for developing targeted interventions.

There is a lack of studies investigating changes in walking performance shortly following a stroke and TIA.[11 12] In the STROKEWALK study, a previously reported randomised controlled trial, we found that individuals receiving text messages for 3 months after stroke and TIA coupled with add-on interventions using a training diary and step counts improved walking distance and chair-rising performance.[13] The present study aims to further study how various baseline characteristics including cardiometabolic risk markers relate to changes in the 6 min walk test, using secondary analyses of data from the STROKEWALK study.

## METHODS
### Study design
This is a post hoc analysis of the STROKEWALK randomised controlled study performed at the Uppsala University Hospital in Sweden. Determination of sample size, recruitment, data collection and random allocation procedures have been previously described in detail.[13] The present study report follows the Strengthening the Reporting of Observational Studies in Epidemiology guidelines for reporting cross-sectional studies[14] and written consent was obtained from all participants.

Recruitment was initiated in November 2016, and the last 3-month follow-up assessment was performed in December 2018. Included were participants aged 18 or above with Transient Ischemic Attack (TIA) or verified stroke (infarction or intracerebral haemorrhage with first or recurrent event) with sufficient cognition (Montreal Cognitive Assessment Scale, MoCA≥26 points), general disability (modified Rankin Scale ≥2) and good enough walking performance; that is, ability to perform the 6 min walking test (with or without a walking aid).[13] The exclusion criteria were known subarachnoid haemorrhage, medical problems such as uncontrolled hypertension, untreated arrhythmias, unstable cardiovascular conditions, a dementia diagnosis, severe aphasia, severe psychiatric problems or cognitive impairment with difficulties understanding instructions.[13]

### Study outcomes
The 6 min walk test was used to measure the maximal walking distance during 6 min over a 30 m course. Changes are described as differences in walking distance at 3 months.[13 15] All baseline data were collected on one occasion while the patients were still treated at the hospital or the first days after discharge and after 3 months close to the end of interventions.

### Baseline assessments
All baseline data were collected on one occasion while the patients were still treated at the hospital or the first days after discharge.

The modified Rankin Scale was used to assess general disability and is scored from 0 (no symptoms) to 6 (dead).[13 16]

Cognitive function at baseline was assessed using the MoCA scale (0–30 points),[13 17] with a higher value indicating better function.

The Charlson Comorbidity Index (CCI) was used to classify comorbid conditions.[13 18]

The last registration of supine blood pressure was recorded manually before discharge from the hospital. Smoking and education levels were assessed by yes or no answers to the questions: 'Are you a smoker at this time of your life?' and 'Do you have a university degree?' For the chair-stand test, the participant was instructed to rise from a seated position without support as quickly as possible five times in a row.[13 19] The test was performed with standardised instructions from the Short Physical Performance Battery.[13 20] The chair-stand test was a measure of lower body strength and the severity of the lower limb impairment.

The 10 m walk test was used to measure comfortable walking speed.[13 21]

From the patient's medical records, cardiometabolic biochemical risk factors such as total, low-density lipoprotein and high-density lipoprotein cholesterol, C reactive protein and triglycerides were analysed at hospital admission and diagnoses such as diabetes, hypertension, hypercholesterolaemia and cardiac heart failure were registered.[22] Biochemical analyses were performed by accredited methods of the Clinical Chemistry Laboratory at the Uppsala University Hospital, Uppsala. Glycated haemoglobin was analysed using non-fasting venous blood samples.

Body mass index (BMI) was calculated as body weight (kg) divided by height (m) squared. Weight was recorded with participants wearing light indoor clothing. Height was measured to the nearest cm. A hip-mounted pedometer at weeks 1 and 12 was used for daily step count by individuals in the short message service (SMS) group (Yamax, SW-200).

### The SMS-intervention group
The SMS-intervention group received daily text SMS (no cost for the participants) as an addition to standard care with simple instructions on what and how to exercise for 3 months. The intervention in the SMS group was composed of three different strategies: (1) 3 months of daily SMS-text messages, (2) training diaries and (3) pedometers for step counts for the first and last week of intervention.[13] The text messages gave instructions

on how to exercise to increase walking endurance and strength of the lower body, without the possibility of texting back for help or advice.

### The control group

Patients in the control group were given standard stroke unit care. They had no restrictions regarding physical activity, exercise or taking part in rehabilitation services and were given standard recommendations. The control group did not use pedometers since it was considered to be a part of the intervention.

The number of individuals who were taking part in rehabilitation services during the study was not recorded.

### Patient and public involvement

A previous pilot study was conducted to test the design of the randomised controlled trial. The intervention was designed in collaboration between individuals with stroke and TIA, healthcare professionals and researchers.

### Statistics

In the STROKEWALK study, a sample size of 80 individuals (including a possible drop-out rate of 20%) was determined to provide secure power to detect a 34 m clinically relevant mean difference in the 6 min walking test.[13] An intention-to-treat analysis was applied for all missing values (drop-outs), and the change was assumed to be 0. Thus, follow-up data for dropouts were registered with a baseline carry-forward approach. Descriptive data are reported as means (SD) and medians (IQR). According to the histogram, normal Q–Q plots, and the Kolmogorov-Smirnov test, data on changes in the 6 min walk test were normally distributed. Case-wise diagnostics and standardised residuals were used to identify potential outliers. Differences in step counts from baseline to 3 months for the SMS group were calculated with the Student's paired-sample t-test. Baseline differences between those that improved ≥34 m or <34 m in the 6 min walk test were assessed using the Student's t-test for continuous, normally distributed variables, and the Mann-Whitney-U test was applied for ordinal or non-normally distributed variables. The $\chi^2$ test was used for categorical variables. The cut-off of 34 m was used for power analyses in the original study.[13]

In the regression analyses, baseline explanatory variables for changes in the 6 min walk test were first identified by correlation and univariate regression analyses (p<0 .05). Correlation strength was calculated using Spearman's r for non-parametric data or a Pearson correlation for continuous normally distributed variables. The identified variables were checked for multicollinearity by correlation analysis and cross-tabulation and if the correlation coefficient was 0.80 or more the variable with the lowest r in relation to the dependent variable was omitted from further regression analysis. The baseline 10 m walk test was omitted from further analyses due to multicollinearity. Multiple linear regression analyses were then conducted with the remaining variables to discover which had the greatest impact on changes in walking performance. The ordinal explanatory variable CCI was dichotomised and grouped into 'no comorbidity' or 'one or more than one comorbidity'. The ordinal explanatory variable MoCA scale was dichotomised and grouped to the cut-off score ≥26 points. Changes in the 6 min walk test were used as the dependent variable. We adjusted for age, sex and comorbidity. Case-wise diagnostics showed that one individual could be considered an outlier; that is, with an increase in the 6 min walk test of 365 m, but were not omitted from further analysis. In this study, a sensitivity analysis with complete-case analyses was also carried out. The univariate and multivariate regression analyses were conducted leaving out subjects that dropped out from the study (n=8).

Statistical significance was set at a p<0.05. The SPSS, V.28, was used for the analyses (SPSS).

## RESULTS

Seventy-nine patients with a mean age of 63.9 (10.4) years, 29 women, mean BMI of 27.5 (4.5) $kg/m^2$ were enrolled and allocated to either SMS intervention (n=40) or control group (n=39).[13] Assessments were performed with a median of five (IQR 6) days after stroke or TIA and after 3 months. At baseline assessments, seven individuals temporarily used a walking aid. Seventy-one individuals remained in the study at 3 months and eight individuals had dropped out.

Table 1 gives the clinical characteristics for all individuals at baseline and changes in the 6 min walk test.

At baseline, 27% of participants had a BMI≥30 $kg/m^2$ (obesity), 43% had a BMI between 25 and 29.9 (overweight) and 30% had a BMI<25 $kg/m^2$. In this study, all participants could perform the 6 min walk test on both occasions and no adverse advents occurred during testing. At baseline, the median (IQR) 6 min walk test was 478 (141) m. At 3 months, the median 6 min walk test was 538 (158) m. The median (IQR) change in the 6 min walk test after 3 months was 57 (63) and 23 (73) m for the SMS and control groups, respectively, and the SMS group showed a significantly greater increase in walking distance compared with the control group (p=0.037).[13] On average, the participants in the SMS group walked 6335 steps per day in the first week of intervention and 8173 steps per day after 3 months (p<0.001), an increase of 22.5% (n=33).

### Linear regression analyses for the identification of factors related to changes in walking capacity

Table 2 shows correlations of possible variables for the regression models.

The differences in walking performance were significantly associated in univariate analysis with baseline BMI and the 6 min walk test at baseline (table 3). After adjusting for age, sex and comorbidity, the final model still included a baseline 6 min walk test and BMI, which together with age, sex and no comorbidity explained 36% of the variance (table 3). Younger individuals, men

**Table 1** Baseline characteristics of all patients in the study with ≥34 m increase in the 6 min walking test at 3 months versus patients with <34 m increase in the 6 min walking test

| | Study population | Changes in the 6 min walking test | | | Missing |
| --- | --- | --- | --- | --- | --- |
| | Baseline | ≥34 m (n=38) | <34 m (n=41) | P value | values, n (%) |
| Age, mean (SD)* | 63.9 (10.4) | 61.9 (9.1) | 65.7 (11.2) | 0.01 | 0 |
| Female, n (%)† | 29 (36.7) | 11 (28.9) | 18 (43.9) | 0.17 | 0 |
| SMS group, n (%)† | 40 (50.6) | 26 (68.4) | 14 (34.1) | 0.002 | 0 |
| Control group, n (%)† | 39 (49.4) | 12 (31.6) | 27 (65.9) | | |
| modified Rankin Scale, 0–2† | | | | 0.47 | 0 |
| 0 | 11 (13.9) | 6 (15.8) | 5 (12.2) | | |
| 1 | 53 (67.1) | 23 (60.5) | 30 (73.2) | | |
| 2 | 15 (19.0) | 9 (23.7) | 6 (14.6) | | |
| Diagnosis, n (%)† | | | | 0.70 | 0 |
| Cerebral infarction | 57 (83.5) | 28 (73.7) | 29 (70.7) | | |
| Intracerebral haemorrhage | 9 (11.4) | 5 (13.2) | 4 (9.8) | | |
| TIA | 13 (16.5) | 5 (13.2) | 8 (19.5) | | |
| Charlson Comorbidity Index, n (%)† | | | | | 0 |
| No comorbidity | 47 (59.5) | 25 (65.8) | 22 (53.7) | 0.27 | |
| ≥1 | 32 (40.5) | 13 (34.2) | 19 (46.3) | | |
| BMI, mean (SD)‡ | 27.51 (4.5) | 26.56 (3.77) | 28.39 (5.03) | 0.07 | 0 |
| Diabetes mellitus-2, yes n (%)† | 12 (15.2) | 5 (13.2) | 7 (17.1) | 0.63 | 0 |
| University studies, yes n (%)† | 40 (50.6) | 16 (42.1) | 24 (58.5) | 0.14 | 0 |
| Non-smoking, n (%)† | 71 (89.9) | 33 (86.8) | 38 (92.7) | 0.39 | 0 |
| Step counts (SMS group), mean (SD)‡ | 6335 (2747) | 6612 (2741) | 5757 (2786) | 0.38 | 3 |
| SGPALS, n (%)† | | | | 0.47 | 0 |
| Sedentary | 11 (13.9) | 6 (15.8) | 5 (12.2) | | |
| Light physical activity | 53 (67.1) | 23 (60.5) | 30 (73.2) | | |
| Moderate/high physical activity | 15 (19.0) | 9 (23.7) | 6 (14.6) | | |
| P-HDL cholesterol, (mmol/L), mean (SD)* | 1.32 (0.39) | 1.35 (0.45) | 1.28 (0.34) | 0.44 | 3 (3.8) |
| P-LDL cholesterol, (mmol/L), mean (SD)* | 3.16 (1.09) | 3.18 (1.16) | 3.13 (1.03) | 0.86 | 3 (3.8) |
| P-cholesterol, (mmol/L), mean (SD)* | 5.13 (1.20) | 5.13 (1.30) | 5.13 (1.12) | 0.86 | 3 (3.8) |
| P-triglycerides, (mmol/L), mean (SD)* | 1.35 (0.63) | 1.20 (0.46) | 1.49 (0.73) | 0.03 | 3 (3.8) |
| P-HbA1C, mmol/mol), mean (SD)* | 38.0 (7.54) | 37.74 (8.33) | 38.26 (6.77) | 0.47 | 3 (3.8) |
| P-C reactive protein (mg/L), mean (SD)* | 4.92 (19.40) | 6.80 (27.80) | 3.20 (3.95) | 0.20 | 6 (7.6) |
| P-creatinine (mmol/L), mean (SD)‡ | 85.12 (25.53) | 78.97 (17.83) | 90.80 (30.14) | 0.042 | 2 (2.6) |
| SBP, (mm Hg), mean (SD)‡ | 130.63 (16.56) | 129.83 (16.17) | 131.36 (17.09) | 0.69 | 4 (5) |
| DBP, (mm Hg), mean (SD)* | 76.08 (11.0) | 79.17 (11.24) | 73.23 (10.09) | 0.02 | 4 (5) |

*The Mann-Whitney U test was applied for ordinal or non-normally distributed variables.
†The $\chi^2$ test was used for categorical variables.
‡The Student's t-test was used for continuous, normally distributed variables.
BMI, body mass index; DBP, diastolic blood pressure; HbA1C, glycated haemoglobin; HDL, high-density lipoprotein; LDL, low-density lipoprotein; SBP, systolic blood pressure; SGPALS, Saltin Grimby Physical Activity Scale; SMS, short message service group.

and those with no comorbidity, lower baseline BMI and shorter 6 min walk tests at baseline were more likely to improve their walking performance.

The sensitivity analyses showed that BMI was no longer a strong predictor for changes in the 6 min walk test in the complete-case analyses (n=71). Thus, complete-case analyses are presented in table 4.

**DISCUSSION**

In this post hoc study, we showed in regression analyses that younger age, male sex, no comorbidity, lower baseline BMI and less distance walked in 6 min walk test significantly predicted positive change in 6 min walk test 3 months after stroke or TIA.

**Table 2** Correlation of the variables used in the regression models and changes in walking performance

| | Change 6 min walking test, R | P value |
|---|---|---|
| Age (years)* | −0.39 | <0.001 |
| Sex (female)* | −0.13 | 0.26 |
| CCI (≥1 comorbidity)* | −0.20 | 0.08 |
| BMI, (kg/m$^2$)† | −0.23 | 0.046 |
| 6 min walking test, baseline (metres)† | −0.38 | <0.001 |
| Chair-stand test, (seconds)† | 0.10 | 0.39 |
| Montreal Cognitive Assessment Scale, (≥26 points),* | 0.014 | 0.91 |
| Saltin Grimby Physical Activity Level Scale* (Sedentary, light physical activity, moderate/high physical activity) | 0.078 | 0.49 |

*Correlation strength was calculated using Pearson correlation for continuous and normally distributed variables.
†Correlation strength was calculated using Spearman's r for ordinal and non-normally distributed variables.
BMI, body mass index; CCI, Charlson Comorbidity Index.

In our study sample, those with higher age improved less in the 6 min walk test at 3 months. Thus, our finding is in line with a general tendency to be less physically active at older ages.[14] Age-related physiological changes like reduced oxygen uptake capacity ($VO_2$ max), changes in heart and lungs, loss of muscle mass and strength, more comorbidity, and medications in older age may affect the intensity and ability to perform outdoor walking in the present study.[1 23] However, each individual in this study could find their own suggested intensity level by using the Borg scale.

In the present study, about 70% of the participants had a BMI above 25 kg/m$^2$. Higher BMI predicted less improvement in walking distance as measured with the 6 min walk test. It can be speculated that those with obesity also were sarcopenic; that is, displayed sarcopenic obesity, which is known to affect walking performance.[23] In our study sample, we found weight fluctuation in both directions after 3 months, which might have affected the results.[22] A deterioration in health might be seen in individuals with high BMI due to difficulties being active in daily living. This study indicates that individuals after a stroke and with obesity need help initiating lifestyle changes to increase physical activity.

In a longitudinal study of cardiovascular disease secondary prevention, an inverse association between walking speed and mortality was found; with a 53% reduction in mortality risk for those with the highest walking speed (3.8–6.2 km/hour).[24] In contrast to the present study, the participants were all women, but they had a similar risk factor profile.[24] Another cross-sectional study, using a population 3 months or longer after stroke found that balance as measured with the Berg Balance Scale was a significant predictor of free-living walking activity, and explained 13% of the variance.[12] In our selected sample of individuals with high motor functions at baseline, balance was not a major problem.[13]

Fini *et al* reported a mean of 4078 steps per day, 6 months or more after stroke.[25] The number of steps we found after the first week of SMS intervention in our sample with recent stroke and TIA is higher than that, averaging 6335 steps per day the first week of intervention and a further increase of 22.5% after 3 months. In older adults, a dose–response relationship has been observed for sedentary behaviour, as well as between steps per day and mortality.[26] Walking speed, steps and distance can reflect functional status and health and are important for activities and community ambulation in daily life.[21] Furthermore, this study sample included more males which could have an impact on the results since male sex was associated with greater improvement in walking performance. In contrast to this study with individuals walking independently at the study start, gender did not affect the outcome of a larger study

**Table 3** Univariate and multivariate regression analyses with changes in walking performance (6 min walking test) as the dependent variable and age, sex, comorbidity, baseline BMI and baseline 6 min walk test as independent variables in individuals after stroke and Transient Ischemic Attack (TIA).

| | Univariate analysis | | | | Multivariate analysis | | |
|---|---|---|---|---|---|---|---|
| | Beta standardised | Adjusted R$^2$ | 95% CI | P value | Beta standardised | 95% CI | P value |
| Age, yrs | −0.2 | 0.029 | −3.1 to 0.1 | 0.073 | −0.33 | −3.8 to −1.0 | <0.001 |
| Sex, female | −0.12 | 0.002 | −54.4 to 16.4 | 0.29 | −0.24 | −66.9 to −8.0 | 0.014 |
| CCI, ≥1 comorbidity | −0.19 | 0.024 | −63.8 to 4.9 | 0.09 | −0.16 | −55.5 to 5.4 | 0.11 |
| BMI, kg/m$^2$ | −0.23 | 0.038 | −7.5 to −0.07 | 0.046 | −0.29 | −8.1 to −1.6 | 0.004 |
| 6 min walk test, metres | −0.38 | 0.134 | −0.4 to −0.1 | <0.001 | −0.55 | −0.5 to −0.3 | <0.001 |

The adjusted R$^2$ for the multivariate analysis is 0.36.
An intention-to-treat analysis was performed with drop-outs included (n=8).
BMI, body mass index; CCI, Charlson Comorbidity Index.

**Table 4** Sensitivity analysis: univariate and multivariate regression analyses with changes in walking performance (6 min walking test) as the dependent variable and age, sex, comorbidity, baseline BMI and baseline 6 min walk test as independent variables in individuals after stroke and Transient Ischemic Attack (TIA).

| | Univariate analysis | | | | Multivariate analysis | | |
|---|---|---|---|---|---|---|---|
| | Beta standardised | Adjusted $R^2$ | 95% CI | P value | Beta standardised | 95% CI | P value |
| Age, years | −0.25 | 0.05 | −3.9 to −0.2 | 0.03 | −0.33 | −4.3 to −1.1 | 0.002 |
| Sex, female | −0.11 | −0.002 | −57.5 to 20.8 | 0.35 | −0.24 | −72.7 to −7.1 | 0.018 |
| CCI, ≥1 comorbidity | −0.18 | 0.018 | −66.6 to 9.7 | 0.14 | −0.22 | −67.4 to −2.6 | 0.035 |
| BMI, kg/m$^2$ | −0.21 | 0.031 | −8.1 to 0.43 | 0.08 | | | |
| 6 min walk test, metres | −0.42 | 0.16 | −0.47 to −0.15 | <0.001 | −0.55 | −0.56 to −0.26 | <0.001 |

The adjusted $R^2$ for the multivariate analysis is 0.34.
In the sensitivity analysis, individuals with missing data at 3 months were not analysed (n=8).
BMI, body mass index; CCI, Charlson Comorbidity Index.

(AVERT trial) examining factors associated with walking recovery poststroke.[27]

The reason why those who walked shorter distances at baseline improved the most is unknown, but regression toward the mean cannot be excluded. Still, in those with the poorest walking performance, a smaller increase in steps may be sufficient to have a positive impact on health.[28] New epidemiological studies measuring physical activity with accelerometers show that the positive effects of physical activity may have been underestimated.[28] A large prospective study indicated that up to 10 000 steps a day were associated with a lower risk of cardiovascular incidence and mortality.[28] Additional risk reduction was also found with steps performed at a higher intensity and there was no minimum threshold for the association between increasing steps per day with morbidity and mortality.[28] This can be used to motivate the least active individuals to increase their outdoor walking and number of steps per day.

### Limitations and strengths

Some methodological issues need to be addressed in this study. One limitation is that the study design calls for precaution with causal inferences. Another limitation is that our findings cannot be generalised to more disabled community-living individuals after a stroke or to individuals with chronic stroke. However, since we included participants at the hospital with different socioeconomic statuses and educational backgrounds, we believe this sample to be representative of the acute stroke and TIA population with fewer motor deficits. Finally, the study is relatively small, making the study prone to bias, and all patients are from a single centre in Sweden, therefore, reducing the generalisability of the results.

One strength of the present study is that study data were drawn from a randomised controlled trial and that we used established outcome measures.

### CONCLUSION

In summary, younger age, male sex, no comorbidity, lower BMI and shorter 6 min walk test at baseline contributed most to improvement in walking performance in patients with a recent stroke or TIA. These factors may be important when planning secondary prevention actions. Cost-effective and easily delivered interventions for individuals with minor stroke or TIA to increase walking distance still require further targeted research.

**Author affiliations**
[1]Department of Public Health and Caring Sciences, Geriatrics, Uppsala University, Uppsala, Sweden
[2]Centre for Clinical Research, Uppsala University, Uppsala, Sweden
[3]Department of Public Health and Caring Sciences, Clinical Nutrition and Metabolism, Uppsala University, Uppsala, Sweden
[4]Department of Medical Sciences, Neurology, Uppsala University, Uppsala, Sweden

**Contributors** This study was conceived, organised and managed by BV, SE, UH and EL. BV acts as a guarantor of the study. All authors listed above contributed to the study design and data interpretation. Writing of the first draft of the paper was done by BV and all authors were involved in preparation and critique of the manuscript and reviewed the paper prior to submission.

**Funding** This study was funded by the Medical Faculty at Uppsala University (grant number N/A), the Swedish National Stroke Association (Strokeförbundet) (grant number N/A) and the Geriatric Funding, Sweden (grant number N/A).

**Competing interests** None declared.

**Patient and public involvement** Patients and/or the public were involved in the design, or conduct, or reporting, or dissemination plans of this research. Refer to the Methods section for further details.

**Patient consent for publication** Not applicable.

**Ethics approval** This study involves human participants and was approved by the regional Ethical Review Board of Uppsala University Hospital, Sweden: Dnr: 2015/550. Participants gave informed consent to participate in the study before taking part.

**Provenance and peer review** Not commissioned; externally peer reviewed.

**Data availability statement** Data are available on reasonable request.

and indication of whether changes were made. See: https://creativecommons.org/licenses/by/4.0/.

## ORCID iDs

Birgit Maria Vahlberg http://orcid.org/0000-0002-1508-1435
Staffan Eriksson http://orcid.org/0000-0003-3572-8918
Ulf Holmbäck http://orcid.org/0000-0002-0764-5878

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
