## [Reviewer comments · BMJ Open]

ARTICLE DETAILS

TITLE (PROVISIONAL)	Factors associated with changes in walking performance in individuals three months after stroke or TIA—secondary analyses from a randomized controlled trial of SMS-delivered training instructions in Sweden
AUTHORS	Vahlberg, Birgit; Eriksson, Staffan; Holmbäck, Ulf; Lundström, Erik

VERSION 1 – REVIEW

REVIEWER	Van Bladel, Anke Ghent University
REVIEW RETURNED	08-Aug-2023

GENERAL COMMENTS	This paper reports the results of a post-hoc analysis of a randomized controlled study. They aim to investigate which factors at admission are related to changes in walking performance 3 months post-stroke. Although the subject is of interest, the manuscript at its current form is of insufficient quality to publish. Major concerns: Introduction • The introduction should be rewritten. It insufficiently justifies the research question. There are studies available in literature that have investigated similar research questions. However, these are not discussed in the introduction. Therefore, it is not clear how this study contributes to gaps in the existing literature. Methods • The major concern is to include persons after a TIA. According to the definition of a TIA the symptoms disappear after maximum 24 hours. Therefore, these participants do not have any gait problems anymore at 3 months post stroke. This greatly affects the results of this study and these participants usually do not have the need to be included in rehabilitation.• Also, the inclusion of allocation to the SMS intervention group in the linear regression model is not methodological correct since it was not the aim of this paper to investigate the effect of the SMS intervention, and this is a factor that cannot be generalized to other stroke population not included in this specific intervention group.• I miss an outcome measure specifically related to the severity of the lower limb impairment. This probably contributes for a great part to the change in walking performance but has not been described. Conclusion
--

	 • It is contradictory to write that admission to the SMS intervention group contributes to changes in walking performance and that this information is important when planning SMS rehabilitation services. The effectivity of the intervention is reported in a different paper and should not be included in the regression analyses. • It is reported in the conclusion that “comorbidity” contributes to the improvement in walking capacity. You have to be more specific, so readers understand better what you mean with this. Additional questions:  • Did all participants receive rehabilitation until the measurement point at three months post stroke? • The methods do not describe the moments at which the variables were assessed. This is only given in the results. What was the plan for the first assessment? • What is the p-value at p 10 line 202? Is it the comparison of the change in walking distance between the SMS and the control group? So, there was no significant difference in the change scores? • What is the p value of the change in steps per day for the intervention group? If you say that increased, you must give the p value. However, I don't see the value of reporting the increase in steps per day related to the research question of this paper. Minor corrections: Correct the last sentence on page 8. This does not read fluently. In the title of table 2 (page 10) you use walking capacity and before you used walking performance. Better be consistent. In the statistical analysis you mention that correlations were calculated using a Spearman's rho or Pearson correlation. Table 2 describes the results (r and p values) but does not indicate which correlation coefficient has been calculated. P12 Line 241-246: You mention two times the same sentence. Table 1 needs to include which test has been performed to investigate the group differences. I think the 0.47 of SGPALS is reported in the wrong cell. Table 2 has no legend.
--	---

REVIEWER	Kato, Bernet New England Research Institutes (NERI), Research Scientist / Biostatistician
REVIEW RETURNED	26-Oct-2023

GENERAL COMMENTS	General Comments The paper reports the results of an analysis to investigate factors related to changes in walking performance in individuals who have had a stroke or TIA. The results are of interest for clinicians and public health. Next I present comments on sections of the paper.
--

	Statistics  Line 157: An ITT analysis was applied for all missing values (dropouts) and the change in 6-minute from baseline to 3-months was assumed to be zero. How many subjects dropped out of the study, i.e., subjects that are missing data for some of the variables at 3 months? Did any of the variables used in the multivariate analysis have missing values? If so, even though the sample size is small it would be good to conduct a sensitivity analysis that involves conducting the multivariate regression analysis leaving out subjects that are missing data on some of the variables. Line 163: The Chi-square test, Mann-Whitney U-test, and the student's independent test were used to examine baseline differences ...". For which variables was the Mann-Whitney U-test used? Line 168: For the variables listed in Table 2, correlation strength was calculated using Spearman's rho or a Pearson correlation. Please clarify variables where Spearman's rho was used and where a Pearson correlation was used. Line 173. The base line 10-meter walk test was omitted from further analyses due to multi-collinearity. Does this mean that at baseline data was collected on the 6-meter walk test and the 10-minute meter walk test? Results  Table 1. Clinically what is the significance of comparing baseline characteristics between those for whom the change from baseline to 3-month in the 6-minute walk test was ≥ 34 versus < 34? Table 2. Please add footnotes to clarify the meaning of "*" and "***" on some of the variables in the table. Line 199. The mean 6MWT was 480 (105) meters. Is this the mean and standard deviation of the 6MWT at baseline? Line 200. The mean change in 6-minute walk test was not significantly different between the SMS and control groups (p-value= 0.2). This suggests that the intervention is not beneficial? Is this result consistent with the results reported in the Clinical Rehabilitation (2020) manuscript? Edits  Replace "6MWT" with "6-minute walk test". Line 241 under "Discussion" the sentence beginning "In a longitudinal study of cardiovascular disease ..." appears twice. Please delete one of the sentences. Line 163: The Chi-square test, Mann-Whitney U-test, and the student's independent test were used to examine baseline differences ...". Replace "student's independent test" with "student's t-test".
--	--

VERSION 1 – AUTHOR RESPONSE

Reviewer 1 Dr. Anke Van Bladel	
The introduction should be rewritten. It insufficiently justifies the research question. There are studies available in the literature that have investigated similar research questions. However, these are not discussed in the	Thank you, we have completed this and rewritten the introduction to better match the research question, we added a sentence and 2 references.

introduction. Therefore, it is not clear how this study contributes to gaps in the existing literature.	Walking ability and better balance are associated with higher PA levels in everyday life after stroke and lower mood is related to low PA in people with chronic stroke (6, 7). (page 1, lines 85-87) Se further information in the next comment
The major concern is to include persons after a TIA. According to the definition of a TIA, the symptoms disappear after maximum 24 hours. Therefore, these participants do not have any gait problems anymore at 3 months post-stroke. This greatly affects the results of this study and these participants usually do not have the need to be included in rehabilitation.	Thank you, we have clarified this in the introduction. “Hence, in the work with secondary prevention, actions to increase walking performance soon after a stroke or TIA are needed, and, in this work knowledge about factors associated with changes in walking distance is important (8). Both stroke and TIA indicate ongoing arteriosclerotic changes in the vessels that can lead to further cardiovascular events and PA is known to decrease the risk of stroke, TIA, and myocardial infarction (9-11). There is a lack of studies investigating changes in walking performance in high-functioning individuals soon after stroke and TIA, and the few studies tend to be conducted months after stroke with a narrow focus on physical functions, overlooking possibly important areas such as cognition and cardiometabolic risk markers (8, 12). (page 1, lines 84-95)
Also, the inclusion of allocation to the SMS intervention group in the linear regression model is not methodological correct since it was not the aim of this paper to investigate the effect of the SMS intervention, and this is a factor that cannot be generalized to other stroke population not included in this specific intervention group.	Thanks for the comment. We have performed new statistical analyses without group allocation included in the regression analyses, see Table 3.
I miss an outcome measure specifically related to the severity of the lower limb impairment. This probably contributes for a great part to the change in walking performance but has not been described.	We have clarified that the chair-stand test can be seen as a measure of lower limb impairment in this study. See pages 3-4, lines 141-143.
It is contradictory to write that admission to the SMS intervention group contributes to changes in walking performance and that this information is important when planning SMS rehabilitation services. The effectivity of the intervention is reported in a different paper and should not be included in the regression analyses.	We have in the abstract and at the end of the manuscript removed the conclusion that “admission to the SMS intervention group contributes to changes in walking performance and that this information is important when planning SMS rehabilitation services”. Instead, we added a sentence about the importance of secondary prevention actions: These factors may be important when planning secondary prevention actions. (conclusion, abstract)
It is reported in the conclusion that “comorbidity” contributes to the improvement in walking capacity. You have to be more specific, so readers understand better what you mean with this.	Done, we have expanded the description with “possible no comorbidity” in the conclusion part of the abstract and at the end of the paper.
Did all participants receive rehabilitation until the measurement point at three months post stroke?	We have clarified this in the results by adding another sentence: “Only a few included individuals did receive rehabilitation until the measurement point of three months”, see page 7, lines 216-217.

The methods do not describe the moments at which the variables were assessed. This is only given in the results. What was the plan for the first assessment?	We have clarified baseline assessments (heading) in the Methods section. See page 3, line 129. The 6-minute walk test is described as an outcome assessment, see page 3, lines 126-128.
What is the p-value at p 10 line 202? Is it the comparison of the change in walking distance between the SMS and the control group? So, there was no significant difference in the change scores?	We changed this and now report non-parametric statistics with a significant difference between groups, P=0.037, see page 7, line 231.
What is the p value of the change in steps per day for the intervention group? If you say that increased, you must give the p value. However, I don't see the value of reporting the increase in steps per day related to the research question of this paper.	Thanks. P-value is now reported for the increase in steps. We believe this is important to report, as a measure of compliance. We therefore report the P-value (Page 7, line 233). We also added a sentence concerning statistics in the methods: "Differences in step-counts from baseline to three months for the SMS group was calculated with the Student's paired-sample t-test" . See page 5, lines: 182-183.
Correct the last sentence on page 8. This does not read fluently.	We are not sure which sentence should be corrected. We corrected a sentence in the Discussion "In contrast to this study with individuals walking independently at study start, gender did not affect the outcome of a larger study (AVERT trial) examining factors associated with time to walking recovery post-stroke" , see page 10, lines 300-303.
In the title of table 2 (page 10) you use walking capacity and before you used walking performance. Better be consistent.	We have changed walking capacity -> walking performance, see Table 2.
In the statistical analysis you mention that correlations were calculated using a Spearman's rho or Pearson correlation. Table 2 describes the results (r and p values) but does not indicate which correlation coefficient has been calculated.	Thank you. We have added a description of when Spearman's rho or Pearson correlation was used in Table 2. "Correlation strength was calculated using Spearman's rho for non-parametric data or Pearson correlation for continuous normally distributed variables" .
P12 Line 241-246: You mention two times the same sentence.	Thank you. We have deleted one sentences
Table 1 needs to include which test has been performed to investigate the group differences.	We have added a description of the statistical test used in Table 1: The Student's t-test was used for continuous, normally distributed variables, and the Mann Whitney-U test was applied for ordinal or non-normally distributed variables. The Chi-square test was used for categorical variables.
I think the 0.47 of SGPALS is reported in the wrong cell.	We have changed this (Table 1).
Table 2 has no legend.	We have added abbreviations
Reviewer 2 (Bernet Kato)	
1. Line 157: An ITT analysis was applied for all missing values (dropouts) and the change in 6-minute from baseline to 3-months was assumed to be zero. How many subjects dropped out of the study, i.e., subjects that are missing data for some of the variables at 3 months? Did	A sensitivity analysis was performed and is described in the statistics part and also reported in Table 4. We have clarified in Results that "Seventy-one individuals remained in the study at three months and eight individuals had dropped out" , see page 7, lines 217-218.

any of the variables used in the multivariate analysis have missing values? If so, even though the sample size is small it would be good to conduct a sensitivity analysis that involves conducting the multivariate regression analysis leaving out subjects that are missing data on some of the variables.	
2. Line 163: The Chi-square test, Mann-Whitney U-test, and the student's independent test were used to examine baseline differences ...". For which variables was the Mann-Whitney U-test used?	We have expanded the description of which variables the Mann-Whitney U-test was used in the Methods - Statistics. "Baseline differences between those that improved ≥ 34 meters or < 34 meters in the 6-minute walk test were assessed using the Student's t-test for continuous, normally distributed variables, and the Mann Whitney-U test was applied for ordinal or non-normally distributed variables. The Chi-square test was used for categorical variables".
3. Line 168: For the variables listed in Table 2, correlation strength was calculated using Spearman's rho or a Pearson correlation. Please clarify variables where Spearman's rho was used and where a Pearson correlation was used.	Done. "Correlation strength was calculated using Spearman's rho for non-parametric data or a Pearson correlation for continuous normally distributed variables".
4. Line 173. The baseline 10-meter walk test was omitted from further analyses due to multi-collinearity. Does this mean that at baseline data was collected on the 6-meter walk test and the 10-minute meter walk test?	We have clarified when the 10-meter walk test was performed by adding the heading baseline measurements. See page 3, line 129.
5. Table 1. Clinically what is the significance of comparing baseline characteristics between those for whom the change from baseline to 3-month in the 6-minute walk test was ≥ 34 versus < 34?	This was another way to report data, to clarify better or less improvement associated with baseline characteristics. We used the cut-off of 34 meters when performing power analysis. We added a sentence: The cut-off of 34 meters was used for power analyses in the original study (13). (page 5, lines 187-188)
6. Table 2. Please add footnotes to clarify the meaning of "*" and "***" on some of the variables in the table.	We have removed the "*" and "***" descriptions from Table 2.
7. Line 199. The mean 6MWT was 480 (105) meters. Is this the mean and standard deviation of the 6MWT at baseline?	We now report the median differences with IQR as in the paper published in Clinical Rehabilitation. At baseline, the median (IQR) 6-minute walk test was 478 (141) meters. At three months, the median 6-minute walk test was 538 (158) meters. See page 7, lines 222-223
8. Line 200. The mean change in 6-minute walk test was not significantly different between the SMS and control groups (p-value= 0.2). This suggests that the intervention is not beneficial? Is this	We changed and now report between-group differences as in the Clinical Rehabilitation paper, analyzed with the Mann-Whitney-U test. The median (IQR) change in the 6-minute walk test after three months was 57 (63) and 23 (73)

result consistent with the results reported in the Clinical Rehabilitation (2020) manuscript?	meters for the SMS and control groups, respectively (P =0.037). See page 7, lines 229-231.
9. Replace “6MWT” with “6-minute walk test”.	Done.
10. Line 241 under “Discussion” the sentence beginning “In a longitudinal study of cardiovascular disease ...” appears twice. Please delete one of the sentences.	Thanks. We have deleted one sentence.
11. Line 163: The Chi-square test, Mann-Whitney U-test, and the student’s independent test were used to examine baseline differences ...”. Replace “student’s independent test” with “student’s t-test”.	Done.

VERSION 2 – REVIEW

REVIEWER	Van Bladel, Anke Ghent University
REVIEW RETURNED	11-Dec-2023

GENERAL COMMENTS	Despite the efforts of making a revised version, I believe that the manuscript is not yet ready for publication. Although the authors showed their willingness to expand the introduction, the structure and flow towards the research question is not clear to the readers. The transition from the first to the second paragraph is not clear. Why do you want to talk about outdoor walking? You cannot start talking about “outdoor walking” as a means of increasing physical activity without announcing that first. The sentences are often not correctly formulated. e.g. A reviewed showed an average of 4000 steps in the chronic phase of stroke This should be “A review reported that persons in the chronic phase after stroke only take on average 4000 steps a day” Concerning the inclusion of persons after TIA, the authors now clarified that they include this persons because these are in need of secondary prevention. However, the explanation in the introduction is not placed in an logical order in the story that they want to bring. So, the introduction is, at this point, not of sufficient quality for publication. The authors now added a title “baseline assessments” to clarify when outcome measures were assessed in their methods. However, this does not provide sufficient information. When were they taken? The first day after stroke? Within the first week after stroke? Also related to the changes in the six minute walk test. When was it assessed the first time? If the period between the first assessment and the assessment at 3 months for every participant the same? I can read this information in the results section, but it should be described also in the methods section how the protocol was defined.
---

	In the results section the authors added that a majority of the participants did not receive rehabilitation. Define “a majority”. Additionally, I do not understand that if the participants did not need rehabilitation, how can we measure a change in walking capacity? It seems that most of them did not have a problem with walking capacity because they had a TIA or did not have post-stroke impairments which indicated the need for rehabilitation. The authors need to report the results better. E.g. the median change in the 6 minutes walk test after 3 months was 57 en 23 meters for the SMS and control group (p=0.037). Despite my previous question, this p value still seems to relate to in increase of walking distance over time and not related to a group difference. In the point-to-point answers the authors indicate that the p-value indicates a group difference. Than in the results section they have to write e.g. The SMS group showed a significant greater increase in walking distance compared to the control group (p=0.037). Table 1+ 2: please indicate with a symbols which test you used in which variables. Do not write it in the title, this does not give correct information.
--	---

REVIEWER	Kato, Bernet New England Research Institutes (NERI), Research Scientist / Biostatistician
REVIEW RETURNED	15-Jan-2024

GENERAL COMMENTS	Minor edits: The abbreviation "MWT" has been replaced with "minute walk test" in the majority of the paper. However, there are still a few instances where "MWT" still appears e.g. line 201 on page 10, line 241 on page 12. Replace "6MWT" with "6-minute walk test" wherever it appears.
---

VERSION 2 – AUTHOR RESPONSE

Reviewer 1 Dr. Anke Van Bladel	
Although the authors showed their willingness to expand the introduction, the structure and flow towards the research question is not clear to the readers. The transition from the first to the second paragraph is not clear. Why do you want to talk about outdoor walking? You cannot start talking about “outdoor walking” as a means of increasing physical activity without announcing that first. The sentences are often not correctly formulated. e.g. A reviewed showed an average of 4000 steps in the chronic phase of stroke This should be “A review reported that persons in the chronic phase after stroke only take on average 4000 steps a day” Concerning the inclusion of persons after TIA, the authors now clarified that they include this persons because these are in need of	Introduction: We have rewritten the entire introduction to better match the research question. and we now think that both the flow and structure are improved. We believe that the language was also improved throughout the introduction, including the sentence you used to exemplify the poor language. For example, your example sentence is now formulated: “Individuals who have suffered from a stroke take an average of 4000 steps a day in the chronic phase after stroke, which is far below the recommended 10,000 steps to meet the guidelines for physical activity (6, 7).” See page 1, lines 81-84. We have rewritten the introduction to better match the research question and we now think that the flow and the structure are improved. See comments above.

secondary prevention. However, the explanation in the introduction is not placed in an logical order in the story that they want to bring. So, the introduction is, at this point, not of sufficient quality for publication.	
The authors now added a title “baseline assessments” to clarify when outcome measures were assessed in their methods. However, this does not provide sufficient information. When were they taken? The first day after stroke? Within the first week after stroke?	We added a description of when data were collected. “All baseline data were collected on one occasion while the patients were still being treated at the hospital or the first days after discharge”. See page 3, lines 130-132.
Also related to the changes in the six minute walk test. When was it assessed the first time? If the period between the first assessment and the assessment at 3 months for every participant the same?	We have tried to make the text clearer by adding “All baseline data were collected on one occasion while the patients were still being treated at the hospital or the first days after discharge and after three months close to the end of interventions.” See page 3, lines 135-136.
In the results section the authors added that a majority of the participants did not receive rehabilitation. Define “a majority”.	We have deleted the sentence as we do not have exact data on rehabilitation. The deleted information was based on experience from working in the hospital organization for many years, including, at the time of the study. We added a sentence ” The number of individuals that were taking part in rehabilitation services during the study were not recorded”, see page 5, lines 176-177.
Additionally, I do not understand that if the participants did not need rehabilitation, how can we measure a change in walking capacity? It seems that most of them did not have a problem with walking capacity because they had a TIA or did not have post-stroke impairments which indicated the need for rehabilitation.	We hope the changes and the deletion of the sentence concerning rehabilitation are satisfactory. In addition, improvements in the 6-minute walk test can have two different origins. In the study population, a sedentary lifestyle is common but minor motor impairments are also present to some degree. Therefore, the improvements in the 6-minute walk test could be due to improved walking capacity as well as improved aerobic capacity.
E.g. the median change in the 6 minutes walk test after 3 months was 57 en 23 meters for the SMS and control group (p=0.037). Despite my previous question, this p value still seems to relate to in increase of walking distance over time and not related to a group difference. In the point-to-point answers the authors indicate that the p-value indicates a group difference. Than in the results section they have to write e.g. The SMS group showed a significant greater increase in walking distance compared to the control group (p=0.037).	We added the suggested sentence to clarify the results. The median (IQR) change in the 6-minute walk test after three months was 57 (63) and 23 (73) meters for the SMS and control groups, respectively and the SMS group showed a significantly greater increase in walking distance compared to the control group (P =0.037) (13). See page 8, lines 239-240.
Table 1+ 2: please indicate with a symbols which test you used in which variables. Do not write it in the title, this does not give correct information.	To clarify statistics, symbols are now being used to connect to each variable in the tables. When checking statistics, some minor errors in T1-T3 were found and action was taken to correct them. T1: The following corrections are being performed:

	Overall, we decided to present the proportion calculation to match each subgroup (≥ 34 meters or < 34 meters). The number of individuals with cerebral infarction is 57. The mean value for the following variables has been changed: P-HDL, P-cholesterol, P-LDL-cholesterol, P-triglycerides, P-HbA1c, and P-creatinine The p-value for group differences concerning step counts is p= 0.38. The p-value for SBP is p=0.69. We found a missing value of SD for creatinine, which is added (SD=30.14). T2: The p-value and r-value for the chair-stand test have been changed and are r=0.10 and p=0.39. We have also added footnotes to the T1 and T2 to describe which tests that have been used for analysis. The P-value for comorbidity in Table 3 has been changed from 0.12 to 0.11 when performing multiple regression analyses. The standardized Beta-value for age should be - 0.20 in the univariate analysis. The increase in step-counts should be 22.5%. See page 8, line 242
Reviewer 2 Dr Bernet Kato, New England Research Institute	
The abbreviation "MWT" has been replaced with "minute walk test" in the majority of the paper. However, there are still a few instances where "MWT" still appears e.g. line 201 on page 10, line 241 on page 12. Replace "6MWT" with "6-minute walk test" wherever it appears.	The abbreviation "MWT" has been replaced with "6-minute walk test" at all places it appears in the manuscript.